# Better Safe than Sorry: Pre-training CLIP against Targeted Data Poisoning and Back­door Attacks

## Abstract

Contrastive Language-Image Pre-training (CLIP) on large image-caption datasets has achieved remarkable success in zero-shot classification and enabled transfer­ability to new domains. However, CLIP is extremely more vulnerable to targeted data poisoning and backdoor attacks, compared to supervised learning. Perhaps sur­prisingly, poisoning 0.0001% of CLIP pre-training data is enough to make targeted data poisoning attacks successful. This is four orders of magnitude smaller than what is required to poison supervised models. Despite this vulnerability, existing methods are very limited in defending CLIP models during pre-training. In this work, we propose a strong defense, SafeCLIP, to safely pre-train CLIP against targeted data poisoning and backdoor attacks. SafeCLIP warms up the model by applying unimodal contrastive learning (CL) on image and text modalities sepa­rately. Then, it carefully divides the data into safe and risky subsets. SafeCLIP trains on the risky data by applying unimodal CL to image and text modalities separately, and trains on the safe data using the CLIP loss. By gradually increasing the size of the safe subset during the training, SafeCLIP effectively breaks targeted data poisoning and backdoor attacks without harming the CLIP performance. Our extensive experiments show that SafeCLIP decrease the attack success rate of targeted data poisoning attacks from 93.75% to 0% and that of the backdoor attacks from 100% to 0%, without harming the CLIP performance on various datasets.

## 1 Introduction

Pre-training large vision-language models on enormous amount of paired image-caption data crawled from the internet has achieved remarkable success in zero-shot classification and robustness to distribution shift. CLIP learns image and text representations in a shared space by maximizing the agreement between the paired image-text representations, and minimizing the aggreement between the unpaired ones. This alleviates the need for high-quality annotations and allows scaling up the pre-training data to millions (Radford et al., 2021) and billions of examples (Jia et al., 2021). Despite the superior performance, CLIP is extremely vulnerable to targeted data poisoning and backdoor attacks, where an adversary injects a subset of malicious examples in the training data to change the prediction of particular examples at test time. Perhaps surprisingly, poisoning only 0.0001% and 0.01% of the pre-training data is enough to make targeted data poisoning and backdoor attacks successful, respectively (Carlini et al., 2023; Carlini & Terzis, 2021). Considering that the large pre-training data of CLIP is often crawled from the internet, such attacks are very easy to perform in practice.

Despite this vulnerability, protecting CLIP against targeted data poisoning and backdoor attacks during pre-training has remained unaddressed. The only recently proposed method, RoCLIP, aims to disassociate the poisoned image-caption pairs during pre-training by matching the image repre­sentations with the nearest neighbors of their captions, and matching the caption representations with the nearest neighbors of their images (Yang & Mirzasoleiman, 2023). However, this method can only defend CLIP against a relatively small number of poisons. Two other methods proposed to clean a *poisoned pre-trained* CLIP, by fine-tuning on a *clean* data of the same scale as pre-training (Yang et al., 2023), or fine-tuning on a *clean* subset of pre-training data using CL on image and text modalities (Bansal et al., 2023). The first method is clearly not applicable to pre-training, and the

second one even increases the attack success rate if applied to *poisoned* pre-training data, as we will confirm experimentally.

Protecting CLIP against targeted data poisoning and backdoor attacks during pre-training is indeed very challenging. This is because training only once on the poisoned pairs can make the attack successful. In contrast, in the supervised setting the model should be trained on the poisoned data for several epochs before the attack succeeds (Biggio et al., 2012; Turner et al., 2019). Thus, to protect CLIP during pre-training, it is crucial to entirely exclude the poisoned examples from the pre-training pipeline.

In this work, we propose the first effective defense, SAFECLIP, against *strong* targeted data poisoning and backdoor attacks during pre-training CLIP. SAFECLIP first warms up the model by applying unimodal CL on the image and text modalities separately. This initializes the model in a way that poisoned image-caption representations have a low similarity initially. Then, it applies the CLIP loss with a *low learning rate* to associate image-caption representations, while maintaining a low similarity for poisoned pairs. Subsequently, SAFECLIP divides the data into a small safe set and a large risky set based on the similarity of their image-caption representations. SAFECLIP pre-trains the model by applying the CLIP loss only to the safe set and applying CL to the image and text modalities of the risky set separately. The safe and risky sets are updated during the training and the size of the safe set is gradually increased. In doing so, SAFECLIP effectively excludes the vast majority of the poisoned examples from the safe set and prevents CLIP loss to associate their poisoned image-captions. This effectively breaks the attack. SAFECLIP ensures a superior performance by increasing the size of the safe set applying data augmentation to its examples during pre-training.

We conduct extensive experiments on the Conceptual Caption (CC) 1M to evaluate the effectiveness of SAFECLIP. We show that SAFECLIP effectively breaks state-of-the-art targeted data poisoning and backdoor attacks during pre-training, by decreasing the success rate of targeted data poisoning attacks from 93.75% to 0% and that of backdoor attacks from 54.3% to 0%, without harming the zero-shot and linear prob performance of CLIP on various datasets.

## 2 RELATED WORK

**Unimodal Contrastive Learning (CL)** Unimodal contrastive learning is among the most successful methods for representation learning (Chen et al., 2020; Caron et al., 2020; Chen & He, 2021). CL maximizes the agreement between different augmented views of the same example (positive pairs) while minimizing the agreement between augmented views of different examples (negative pairs). A recent body of work aimed to further improve the performance of CL, by improving the consistency of the representations via a momentum encode (He et al., 2020), eliminating the need for negative pairs (Grill et al., 2020), or removing redundancy between components of the representation vectors (Zbontar et al., 2021). Most relevant to our work is NNCLR, which enriches the learned representations by keeping a memory bank of augmented representations and use the nearest neighbor of every example in the pool as its positive pair (Dwibedi et al., 2021).

**Contrastive Language-Image pre-training (CLIP)** Large vision-language models like CLIP (Radford et al., 2021) and ALIGN (Jia et al., 2021) achieved a remarkable success by contrastive pre-training on 400M and 1B image-caption pairs crawled from the web. Recent work tried to improve the data efficiency and performance of CLIP. Specifically, DeCLIP (Li et al., 2021) uses SimSiam (Chen & He, 2021) and Masked Language Modeling (Devlin et al., 2018) to match the augmented views of the image representations and the augmented views of the text representations, to improve the data efficiency of CLIP. SLIP (Mu et al., 2022) improves the performance by including in-modal contrastive learning on images using SimCLR, which maximizes the agreement between different views of the same augmented image while minimizing agreement between augmented views of different images. CyCLIP (Goel et al., 2022) emphasizes the importance of in-modal consistency, where the difference in similarities of image-image pairs should be close to that of text-text pairs, and cross-modal consistency, where the difference in similarities of each image-text pair should be similar.

**Targeted Data Poisoning and Backdoor Attacks on CLIP** CLIP is highly susceptible to various types of targeted data poisoning and backdoor attacks (Carlini & Terzis, 2021; Yang et al., 2023). Targeted data poisoning attacks (TDPA) aim to deceive the model into misclassifying a specific test example by modifying the captions of a small subset of the training data. Backdoor attacks (BA) involve embedding a backdoor trigger into a small subset of examples in the training data, with the goal of causing the model to misclassify any test images with the same trigger. A backdoor trigger

can be either visible, like a distinguishable patch, or invisible, like patterned noise points or patterned image deformation (Chen et al., 2017; Gu et al., 2017; Nguyen & Tran, 2021). Adding trigger to only 0.01% of the pre-training data can cause the model to misclassify the backdoored examples. TPDA is even more effective, requiring only 0.0001% of the data to be poisoned (Carlini & Terzis, 2021).

**Targeted Data Poisoning and Backdoor Defense on CLIP** Despite the vulnerability of CLIP to TDPA and BA, existing defense methods are very limited. RoCLIP (Yang & Mirzasoleiman, 2023) is the only proposed defense for protecting CLIP during pre-training. RoCLIP first augments image-caption pairs using techniques such as random cropping and color jittering. Subsequently, it matches each image with the nearest neighbor of its caption, and vice versa. These nearest-neighbors are drawn from a representation pool, which is updated at the end of every epoch. However, RoCLIP is effective only against a limited range of poisons and fails to defend the model when trained on datasets with a poison rate higher than 0.0015%.

Two recent works proposed data cleansing for fine-tuning CLIP, or cleaning a poisoned pre-trained CLIP during fine-tuning. Yang et al. (2023) proposed dropping examples that have a low image-caption similarity based on a clean pre-trained CLIP, to cleanse the fine-tuning data. This method requires a clean pre-trained model, and a proper threshold to filter the poisons without discarding a large amount of clean data. This threshold varies for different attack types and is difficult to pre-compute. To clean a poisoned CLIP with TDPA, Yang et al. (2023) proposed fine-tuning on a clean dataset of the same size as the pre-training data. Moreover, to clean a poisoned CLIP with BA, Bansal et al. (2023) proposed CleanCLIP, which fine-tunes the model on a *clean* subset of the pre-training data with CLIP loss and CL loss on image and text modalities. The first method is clearly not applicable to pre-training and the second one increases the attack success rate when applied to the poisoned data. This is because CL cluster the backdoored images and their cpations, and the CLIP loss can even better associate the backdoored images with the poisoned captions.

In this work, we propose the first effective defense for protecting CLIP against strong TDPA (0.05%) and BA (0.05%) during pre-training, without compromising the model's performance.

## 3 PRELIMINARY

### 3.1 CONTRASTIVE LANGUAGE-IMAGE PRE-TRAINING (CLIP)

Consider a dataset $\mathcal{D} = \{(\boldsymbol{x}_i^{\mathcal{I}}, \boldsymbol{x}_i^{\mathcal{T}})\}_{i=1}^n$ of $n$ image-captions pairs, where $\boldsymbol{x}_i^{\mathcal{I}}$ and $\boldsymbol{x}_i^{\mathcal{T}}$ are the image and caption of the $i^{th}$ pair. The CLIP architecture consists of an image encoder $f_I : \mathcal{I} \rightarrow \mathbb{R}^d$ and a text encoder $f_T : \mathcal{T} \rightarrow \mathbb{R}^d$ to encode images and captions. The encoded representations are projected into the same space and are normalized to have unit $\ell_2$-norm. We denote the resulting image and text representations by $\boldsymbol{z}_i^{\mathcal{I}}, \boldsymbol{z}_i^{\mathcal{T}}$. To create the multi-modal interaction, the InfoNCE loss is applied to pull the projected representations of every image-caption pair together while pushing apart the projected representations of unpaied images and captions in the same mini-batch. Formally, for a mini-batch of $N$ pairs, the CLIP loss is defined as:

$$\mathcal{L}_{\text{CLIP}} = -\frac{1}{2N} \sum_{j=1}^N \log \left[ \frac{\exp\left(\langle z_j^{\mathcal{I}}, z_j^{\mathcal{T}} \rangle / \tau\right)}{\sum_{k=1}^N \exp\left(\langle z_j^{\mathcal{I}}, z_k^{\mathcal{T}} \rangle / \tau\right)} \right] - \frac{1}{2N} \sum_{k=1}^N \log \left[ \frac{\exp\left(\langle z_k^{\mathcal{I}}, z_k^{\mathcal{T}} \rangle / \tau\right)}{\sum_{j=1}^N \exp\left(\langle z_j^{\mathcal{I}}, z_k^{\mathcal{T}} \rangle / \tau\right)} \right], \quad (1)$$

where $\tau$ is a trainable temperature parameter, and $\langle ., . \rangle$ is the inner product between two representations. The performances of CLIP is evaluated with zero-shot or linear-probe, as we discuss next.

**Zero-shot classification.** Zero-shot classification assess the generalizability and transferability of the model to unseen tasks. It transforms the downstream labels into natural language captions using the provided engineered prompt templates, such as `"A photo of a {label}"` (Radford et al., 2021). Then, it calculates the cosine similarity between the representations of a given image and each prompt, and predicts the label with the highest image-prompt similarity.

**Linear probe classification.** Linear probe classification refers to evaluating the extracted representations from the pre-trained image encoder for training a linear classifier on the downstream labeled data.

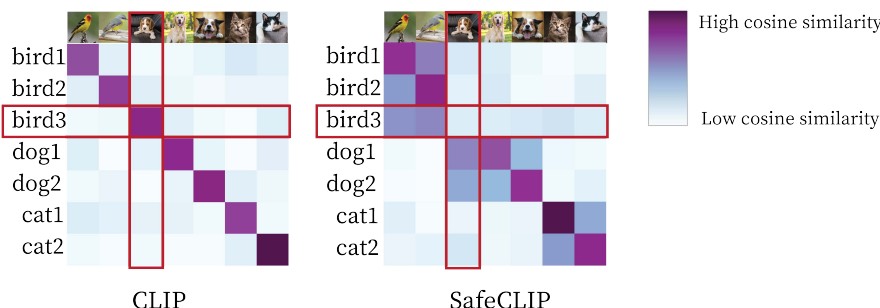

Figure 1: Cosine similarity matrices between image-caption representations, obtained by CLIP and SAFECLIP. Training with CLIP loss on the poisoned data associates the poisoned image-caption pairs. In contrast, SAFECLIP prevents association of the poisoned image-caption pairs, by clustering images and captions in the same category via an inmodal CL loss before pre-training.

### 3.2 TARGETED DATA POISONING AND BACKDOOR ATTACKS

Targeted data poisoning and backdoor attacks poison CLIP by injecting a set of poisoned image-caption pairs to the pre-training data. Let $\mathcal{D}_p = \{(x_i^{\mathcal{I}}, x_c^{\mathcal{T}}) | x_i^{\mathcal{I}} \in \mathcal{I}_t, x_c^{\mathcal{T}} \in \mathcal{T}_{adv}\}$ be the injected poisoned pairs, where $\mathcal{I}_t$ is the poisoned image(s) and $\mathcal{T}_{adv}$ is the set of adversarial caption related to the adversarial label $y_{adv}$. To construct the poisoned caption set, one can search the training dataset for all captions that contain the adversarial label and use these captions as the adversarial captions. Another approach is to use CLIP's set of 80 different prompt-engineered text descriptions (Radford et al., 2021) to construct captions for the adversairal label, and then either use a subset of them or repeat them as necessary. In our work, we construct $\mathcal{T}_{adv}$ from the training dataset, which is consistent with the construction methods used in (Carlini & Terzis, 2021; Yang et al., 2023; Yang & Mirzasoleiman, 2023; Bansal et al., 2023).

**Targeted data poisoning attacks** aim to misclassify a particular test example, $x_i^{\mathcal{I}}$, as $y_{adv}$. Hence, $D_p = \{(x_i^{\mathcal{I}}, x_c^{\mathcal{T}}) | x_c^{T} \in \mathcal{T}_{adv}\}$.

**Backdoor attacks** introduce a trigger patch to a set of poisoned images. The goal is to misclassify any test examples with the trigger patch, $x_i^{\mathcal{I}} \oplus$ patch, as $y_{adv}$. Hence, $D_p = \{(x_i^{\mathcal{I}} \oplus \text{patch}, x_c^{\mathcal{T}}) | x_i^{\mathcal{I}} \in \mathcal{I}, x_c^{T} \in \mathcal{T}_{adv}\}$. In contrast to targeted data poisoning attacks which target a particular test example, backdoor attacks inject *random* images with the backdoor trigger, paired with the adversarial captions.

**Adversary Objective** The primary objective of the adversary is to manipulate the output representations of CLIP, such that certain images are misclassified into adversarial categories instead of their true categories, while the other images are classified correctly.

**Adversary Capabilities** We assume that the adversary has limited control over the pre-training data, and can inject a small number ($\leq 0.05\%$ of the dataset size) of poisoned examples into the training dataset. Adversary also has the knowledge of the model structure, the training algorithm, and the hyperparameter used by their victim, but they cannot modify the training process directly.

## 4 METHOD

Next, we motivate and present SAFECLIP for robust pre-training of CLIP against TDPA and BAs.

### 4.1 MOTIVATION

Targeted data poisoning and backdoor attacks can succeed extremely fast when pre-training CLIP models. For example, when pre-training on a dataset with 0.01% poison rate, only 1 epoch of CLIP is enough to poison the model. Thus, to prevent the model from being poisoned, it is essential to filter out the majority of poisoned pairs *before* the pre-training starts, and keep them out *throughout* the pre-training. If the model avoids training on or is exposed to only a limited amount of the poisoned data, the representations of poisoned images and captions do not get close during pre-training, and the attack fails. However, as shown in Fig. 2, the poisoned pairs become inseparable from the clean pairs after 1 pre-training epochs. To filter out the poisoned pairs, SAFECLIP first warms up the model by a few epochs of unimodal CL on image and text modalities, separately. The unimodal CL clusters simliar images and similar texts together. In doing so, it effectively pushes the poisoned image-caption representations apart, thus slowing down the attacks from taking effect. Then, SAFECLIP runs 1

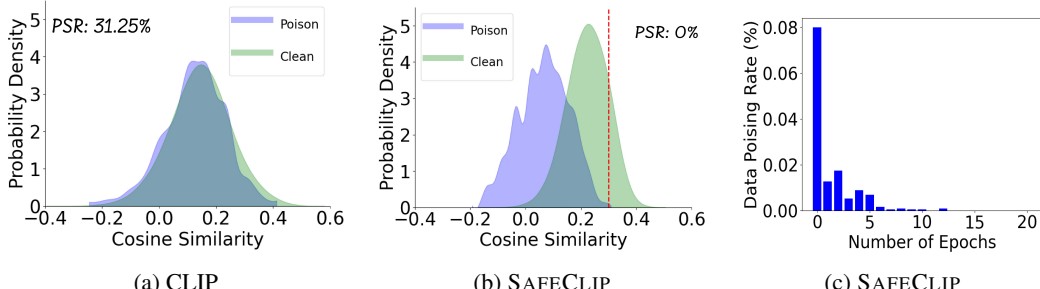

(a) CLIP      (b) SAFECLIP      (c) SAFECLIP

Figure 2: Distribution of image-caption cosine similarity after 1 epoch of pre-training with (a) CLIP and (b) SAFECLIP. The data to the right of the red line represents the portion of the (safe) data used for training with CLIP loss. SAFECLIP almost exclusively applies the CLIP loss to clean data. (c) Fraction of remaining poisons throughout pre-training with SAFECLIP. After the warmup, the poison ratio drops from 0.08% to 0.018%, and gradually goes to 0% as the pre-training continues. This shows the effectiveness of SAFECLIP in filtering out the poisoned pairs and defending the model.

epoch of CLIP with *small learning rate* and evaluates the cosine similarities of all examples. As shown in Fig. 2, after the initial warmup epochs, the poisoned pairs can be better separated from the clean pairs. Since poisoned pairs are less similar compared to clean pairs, SAFECLIP only trains with the CLIP loss on examples with high cosine similarity (safe data), while training on other examples (risky data) with unimodal CL loss. Throughout the pre-training, SAFECLIP gradually increases the amount of data used for training with CLIP loss. Since majority of the poisoned pairs are never trained on, SAFECLIP can successfully defend the model from strong targeted data poisoning and backdoor attacks. Fig. 2c shows that the poison ratio in the pre-training data remains low throughout pre-training with SAFECLIP, while more clean pairs are being used to pre-train with the CLIP loss.

In summary, to prevent the model from being poisoned, SAFECLIP consists of three steps: (1) A few epochs of unimodal CL warmup. (2) One epoch of slow-paced CLIP warmup. (3) A combination of unimodal CL training and CLIP training with data updating. The effect of SAFECLIP on image and text encoders is shown in Fig 1. CLIP focuses on aligning the paired image-caption representations, which renders it easy to poison. On the other hand, SAFECLIP clusters images and captions in the same category. In doing so, it reduces the similarity of poisoned image-caption representations, This allows SAFECLIP to successfully defend strong poisoning and backdoor attacks. The pseudocode of the SAFECLIP is illustrated in Alg. 1. Next, we will discuss each step in more details.

### 4.1.1 UNIMODAL SELF-SUPERVISED WARMUP

SAFECLIP leverages unimodal CL on both image and text modalities separately. Since unimodal CL does not match poisoned images with captions, it does not risk poisoning the models. The unimodal CL clusters similar images and similar captions together. In doing so, poisoned pairs can be better separated from the clean pairs. Effectively, since the poisoned image(s) and adversarial captions are from different categories, poisoned images and adversarial captions cluster with their respective true representation clusters during unimodal CL warmup, and move further away from each other in representation space. For example, to poison an image of "cat" with a "plane" caption, the image needs to move closer to the "plane" text cluster and away from the "cat" image cluster in the representation space. The closer the image is to its true "cat" representation cluster at the beginning of training, the more challenging it becomes to poison the image. Same argument applies to captions.

**Nearest-Neighbors** When the poison rate is high, poisoned images, which are either identical images (TDPA) or images sharing the backdoor patch (BA) cluster tightly together in the representation space. This prevent them from getting close to the cluster of their true category. To avoid this issue and enrich the quality of the representations, we extend our unimodal CL training by using a nearest neighbor pool to find positive pairs (Dwibedi et al., 2021). That is, rather than directly matching differently augmented views of the same image or caption $(z_i, z_i^+)$, we maintain a pool of image and caption representations, and match each image or caption representation with their nearest neighbor in the pool rather than its augmented views. The pool is implemented as a queue, initially initialized with representations of random examples and is updated by including the representations of examples in the current mini-batch, while excluding the oldest representations in the queue. By exposing examples to more diverse positive pairs, SAFECLIP prevents clustering of poisoned images and adversarial captions, and can separate the poisoned pairs more effectively. We will explore the impact

of using the nearest neighbor approach in our ablation study. The unimodal CL loss is defined as:

$$\mathcal{L}_{\text{unimodal\_NN}} = -\log \left[ \frac{\exp\left(\langle \text{NN}(\boldsymbol{z}_i, \mathcal{P}), \boldsymbol{z}_i^+ \rangle / \tau\right)}{\sum_{k=1}^{N} \exp\left(\langle \text{NN}(\boldsymbol{z}_i, \mathcal{P}), \boldsymbol{z}_k^+ \rangle / \tau\right)} \right] \tag{2}$$

where $\boldsymbol{z}_i$ is the output image/text representation and $\boldsymbol{z}_i^+$ is the augmented view of the image/text representation, and $\text{NN}(\boldsymbol{z}_i, \mathcal{P})$ is the nearest neighbor operator defined as:

$$NN(\boldsymbol{z}_i) = \text{argmin}_{\boldsymbol{p} \in \mathcal{P}} \|\boldsymbol{z}_i - \boldsymbol{p}\|_2. \tag{3}$$

**Slow-paced CLIP Warmup Epoch** Although unimodal CL brings similar images and captions closer in the image and text representation spaces, the images and their corresponding captions remain relatively distant from each other. Thus, to associate the image-caption representations and effectively distinguish between poisoned and clean pairs, one epoch of CLIP warmup becomes essential before the filtering step. Thus, following the unimodal CL warmup, we proceed with one additional epoch of training with the CLIP loss. Crucially, to mitigate the risk of poisoning, as the CLIP loss directly matches the image-caption pairs, including the potentially poisoned ones, we employ a lower learning rate. This slow-paced CLIP epoch helps prevent the model from learning poisoned image-caption pairs while enabling SAFECLIP to filter out the majority of poisoned pairs before pre-training.

As shown in Fig 2, the warmup results in a significant separation between poisoned and clean pairs. In addition, as we will discuss in Sec. 5.2, only a few epochs of unimodal CL is sufficient. SAFECLIP applies $r = 5$ epochs of unimodal warmup. We will show that this is not sensitive to tuning and can apply to a wide range of poisons in Sec. 5.1.

### 4.1.2  MIXED TRAINING WITH DATA UPDATING

After the warm-up phase, we evaluate the cosine similarities of all the examples and divide them into the safe and risky sets. The top $k\%$ of the data, characterized by high cosine similarity, is considered *safe* data, while the remaining portion is deemed *risky* data. SAFECLIP applies CLIP loss to the safe data, directly matching images with their captions. To ensure the trustworthiness of the safe data, we select a small value for $k$ (e.g., $k = 15$). On the other hand, instead of discarding the risky data, we continue training on it with unimodal CL. However, two concerns still remain: (1) Some poisoned pairs may still be in the safe data, (2) The model's performance may suffer as the CLIP loss is not applied to majority of the examples. To address these concerns, at the end of each epoch, we assess the cosine similarity of all examples and update $k = k + i$, to chose a larger fraction of the data with highest cosine similarity as the new safe set, with $i$ being a small number (e.g., $i = 1$). To further boost the performance, we apply data augmentation to the examples in the safe set used in the CLIP loss.

With the above update strategy, only a small number of poisoned pairs may temporarily enter the safe data and cannot poison the model. At the same time, more training on clean data with CLIP loss and on risky data with unimodal CL loss allows the model to learn better representations and better distinguish and discard the poisoned pairs during the training. Additionally, since we progressively increase the proportion of safe data during training, by the end of the training, the majority of the data will be part of the safe data and will be trained on with CLIP loss, thereby resolving the performance issue. To reduce the computational cost of updating the safe and risky sets, instead of calculating the cosine similarities of all examples at every epoch, we recompute all the similarities every $m$ epochs (e.g. $m = 5$). In other epochs, we only update the cosine similarity for $s\% > k\%$ of examples with the highest similarities, and update the safe and risky set accordingly.

The loss of the mixed training is defined as:

$$\mathcal{L}_{\text{SAFECLIP}}(\mathcal{D}) = \mathcal{L}_{\text{unimodal\_NN}}(\mathcal{D}_{\text{risky}}) + \mathcal{L}_{\text{CLIP}}(\mathcal{D}_{\text{safe\_aug}}). \tag{4}$$

Note that, during mixed training, we still apply nearest-neighbors for unimodal CL.

## 5  EXPERIMENTS

In this section, we evaluate the effectiveness of SAFECLIP against strong TDPA and BA. We start by introducing the experimental setup. Then we present our main results. Finally, we conduct an ablation study on different components of SAFECLIP.

**Training** We used an open-source implementation of CLIP as our base model. Similar to the setup in (Radford et al., 2021), we utilize a ResNet-50 as the image encoder and a transformer as the

---

**Algorithm 1:** SAFECLIP

---

1 **Input:** Image encoder $f_I$, text encoder $f_T$, image pool $P_I$, text pool $P_T$, unimodal warmup epochs $r$, training epochs $T$, safe set initial size $k$, increment ratio $i$
  **Data:** Dataset of image-caption pairs $\mathcal{D} = \{(x_i^{\mathcal{I}}, x_i^{\mathcal{T}})\}_{i=1}^n$, $\mathcal{X}_I = \{x_i^{\mathcal{I}}\}_{i=1}^n$, $\mathcal{X}_T = \{x_i^{\mathcal{T}}\}_{i=1}^n$
2 **for** $t \leftarrow 1$ **to** $r$ **do**
3 $\quad$ Train $f_I$ with $\mathcal{L}_{\text{unimodal\_NN}}(\mathcal{X}_I, P_I)$ in Eq 2
4 $\quad$ Train $f_T$ with $\mathcal{L}_{\text{unimodal\_NN}}(\mathcal{X}_T, P_T)$ in Eq. 2
5 Update $f_I$, $f_T$ by training on $\mathcal{D}$ with $\mathcal{L}_{CLIP}$ in Eq. 1 using a small learning rate
6 **for** $t \leftarrow 1$ **to** $T$ **do**
7 $\quad$ Sort $\{\langle f_I(x_i^{\mathcal{I}}), f_I(x_i^{\mathcal{T}}) \rangle\}_{i=1}^n$ in a decreasing manner
8 $\quad$ $\mathcal{D}_{\text{safe}} \leftarrow$ top $k\%$ of data according to the above ordering
9 $\quad$ $\mathcal{D}_{\text{risky}} \leftarrow \mathcal{D} \setminus \mathcal{D}_{\text{safe}}$
10 $\quad$ $\mathcal{D}_{\text{safe\_aug}} \leftarrow$ augmented examples in $\mathcal{D}_{\text{safe}}$
11 $\quad$ Train $f_I$, $f_T$ with $\mathcal{L}_{\text{SAFECLIP}}(\mathcal{D}) = \mathcal{L}_{\text{unimodal\_NN}}(\mathcal{D}_{\text{risky}}) + \mathcal{L}_{\text{CLIP}}(\mathcal{D}_{\text{safe\_aug}})$
12 $\quad$ $k \leftarrow k + i$

---

text encoder. Due to computational constraints and consistent with (Yang & Mirzasoleiman, 2023), we used Conceptual Captions 3M (CC3M) dataset as our training dataset (Sharma et al., 2018). In each experiment, the model is trained from scratch for 48 epochs with a batch size of 512, using the AdamW optimizer (Loshchilov & Hutter, 2017).

**Downstream Datasets** To evaluate the downstream performance of our model, we conduct linear probe and zero-shot classifications, as introduced in Sec. 3.1, on 10 widely used datasets (Radford et al., 2021; Li et al., 2021; Yang & Mirzasoleiman, 2023) listed in Table 4.

**Adversarial Attacks** To evaluate the effectiveness of our defense, we consider two different attack baselines: targeted data poisoning attacks (TDPA) (Carlini & Terzis, 2021) and backdoor attacks (BA) with visible patch triggers.

For TPDAs, we randomly select 16 different images from the CC3M validation set as our target images. For each target image, we choose a random class from the ImageNet1K dataset (Deng et al., 2009), and construct an adversarial caption set related to the label as discussed in Sec. 3.2. For each target image, we generated 100 and 500 poisons.

For BA, we randomly select 200 and 500 images from the CC3M pre-training data and apply the backdoor trigger. We use the same backdoor triggers as proposed by (Gu et al., 2017). We choose a random class from the ImageNet1K dataset (Deng et al., 2009) and construct the adversarial caption set related to the label as discussed in Sec. 3.2. Each backdoored image is paired with a random poisoned caption from the adversarial caption set. In our experiment, we used the class "mushroom"

**Defense Baselines** We consider two defense baselines against the TDPA and the BA. First is the only pre-training defense, namely RoCLIP (Yang & Mirzasoleiman, 2023), that maintains a pool of representations and matches every image with the nearest neighbor of its caption in the pool, and vice versa. Second is adaptation of CleanCLIP (Bansal et al., 2023) to pre-training, which applies CL and CLIP loss on all the pre-training examples.

We measure the effectiveness of attacks using the attack success rate. For TPDA, Poison Success Rate (PSR) is the fraction of target images that are classified as the adversarial label. For BA, Backdoor Success Rate (BSR) it is the fraction of test images containing the backdoor triggers that are classified as the adversarial label.

### 5.1 MAIN RESULT

Here, we evaluate the performance of SAFECLIP against TDPA and BA. We compare SAFECLIP with baselines, based on both attack success rates and downstream performance.

**SAFECLIP defense** First, we evaluate the effectiveness of SAFECLIP in breaking TDPA and BA. Table 1 shows that for TPDA, SAFECLIP effectively reduces PSR of CLIP from 93.75% to 0% and 6.25% for 100 and 500 poisons, respectively. For BA, it reduces BSR of CLIP from 54.3% and 100% to 0%. This indicates that SAFECLIP can manage to filter the vast majority of the poisoned pairs during the warmup and throughout the training, and successfully defend CLIP during pre-training.

On the other hand, CleanCLIP and RoCLIP exhibit poor defense performances against such strong attacks, and can even increase the attack success rate. RoCLIP fails because the nearest neighbor

Table 1: Effectiveness of SAFECLIP in defending targeted data poisoning attacks (TPDAs) and backdoor attacks (BAs) vs. baselines. The models are trained on 1M data.

| Model | # Poison (TPDA) | PSR (TPDA) | # Backdoor (BA) | BSR (BA) |
|---|---|---|---|---|
| CLIP | 100 | 93.75% | 200 | 54.33% |
|  | 500 | 93.75% | 500 | 100% |
| SAFECLIP | 100 | 0% | 200 | 0% |
|  | 500 | 6.25% | 500 | 0% |
| RoCLIP | 100 | 93.75% | 200 | 0.33% |
|  | 500 | 75% | 500 | 2.33% |
| CleanCLIP | 100 | 87.5% | 200 | 66.67% |
|  | 500 | 93.75% | 500 | 63.33% |

Table 2: Downstream linear probe and zero-shot (top 1) accuracy of SAFECLIP vs CLIP.

| Method | Task | F102 | Fd101 | I1K | Pet | Cars | Cal101 | C10 | C100 | DTD | Air. |
|---|---|---|---|---|---|---|---|---|---|---|---|
| SAFECLIP | 0-shot | 0.62 | 11.1 | 18.2 | 1.5 | 0.9 | 54.4 | 54.7 | 22.6 | 3.56 | 1.1 |
|  | lin-prb | 99.8 | 53.3 | 34.3 | 58.1 | 21.3 | 81.1 | 78.3 | 54.2 | 62.9 | 26.9 |
| CLIP | 0-shot | 1.0 | 13.0 | 19.4 | 3.3 | 1.2 | 50.8 | 48.2 | 18.9 | 3.7 | 1.0 |
|  | lin-prb | 100 | 54.8 | 33.2 | 58.8 | 18.8 | 80.2 | 77.8 | 54.7 | 58.4 | 28.5 |

of the poisoned image and caption is more likely to be another poisoned image and caption when the number of poisons is overwhelming. Similarly, CleanCLIP fails as it applies CLIP loss to all examples including the poisoned pairs. Importantly, this confirms that CL loss is not effective when applied with the CLIP loss to the poisoned data. We see that CleanCLIP increases BSR from 54% to 66.67%, when 200 backdoored images exist in the pre-training data.

**SAFECLIP downstream performance** Next, we evaluate SAFECLIP in terms of the downstream performance, by reporting linear probe and zero-shot performance on 10 downstream datasets in Table 4. As shown in Table 1, SAFECLIP provides a comparable or even superior performance compared to CLIP, which is due to the increasing size of the safe set and using data augmentation.

## 5.2 SAFECLIP WARMUP: ABLATION STUDY AND SENSITIVITY ANALYSIS

SAFECLIP's warmup consist of two essential components: unimodal warmup and a slow-paced CLIP warmup epoch. As introduced in Sec. 4.1.1, during unimodal warmup, SAFECLIP applies CL to image and text modalities separately, while during slow-paced CLIP warmup, SAFECLIP applies 1 epoch of CLIP loss to all examples with a lower learning rate. Here, we illustrate the necessity of each of these components. We randomly select 16 different images as our target images, and each target image is paired with 100 poisoned captions related to the adversarial labels. In total, there are 1,600 posioned pairs injected into the dataset.

**Unimodal CL Warmup** Unimodal CL is essential to the success of SAFECLIP. As shown in Table 3, row 1,2: 5 epochs of unimodal CL warmup enables the model to filter out 7% more of the total poisons from the top 30% of examples with highest cosine similarities, compared to 1 epoch of unimodal CL. However, as shown in row 3, the benefit of unimodal CL is rather diminishing, and 5 more epochs of the unimodal training before filtering can only filter out 0.1 % more poisons in the top 30%.

Table 3: Effect of number of CL and slow-paced CLIP warmup epochs on poison rate (PSR)

| # CL epochs | # CLIP epochs | PSR in top 30% |
|---|---|---|
| 5 | 1 | 0.8% |
| 1 | 1 | 7.8% |
| 10 | 1 | 0.7% |
| 5 | 0 | 27.5% |
| 5 | 2 | 6.8% |

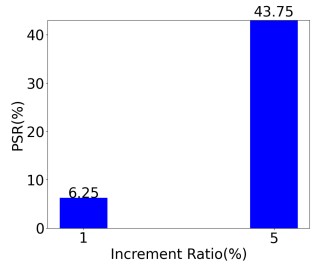

Figure 4: Effect of using low increment ratio when updating $\mathcal{D}_{\text{safe}}$

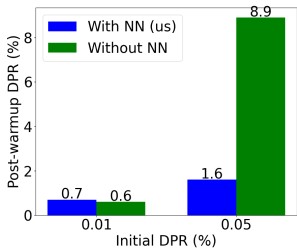 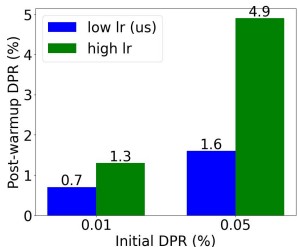 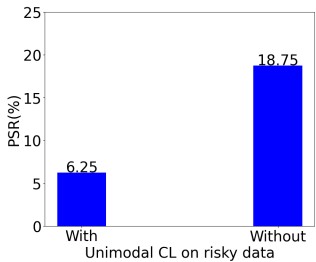

(a) Effect of nearest-neighbors     (b) Effect of low learning rate     (c) Effect of unimodal CL on $\mathcal{D}_{\text{risky}}$

Figure 3: Effect of using (a) near-neighbors during unimodal CL warmup and (b) low learning rate during CLIP warmup. (c) unimodal CL on *risky* data during mixed training. (a) and (b) show the remaining fraction of poison in the top 30% data after the warmup, for various initial poison rates, while (c) shows the PSR at the end of the training.

**Slow-paced CLIP warmup** Although CLIP training on full data exposed the model to the poisoned pairs, to correlate the image representation with the caption representation and better filtering the poisons out, it is essential to train with CLIP loss for 1 epoch at a *low* learning rate. As shown in Table 3, row 1,4: without any CLIP training, there are 26.7% more poisons in the top 30% of training data. At the same time, it is crucial to not train more with CLIP loss on the full data before filtering out the poisoned pairs. As shown in row 5, applying 1 more CLIP epoch with low learning rate to the poisoned data introduces 6% more poisons in the top 30%.

In addition, it is important to keep the learning rate low during the CLIP warmup. As shown in Fig. 3b, with a lower lr, 0.9% less poisons are found in the top 30% when the dataset has a 0.01% poison rate, and 3.3% less poisons are found in the top 30% when the dataset has a 0.05% poison rate.

**Ablation on nearest-neighbors** To study the effect of using nearest-neighbors in the unimodal CL warmup, we randomly select 16 different images as target, and each target image is paired with 100 or 500 captions related to the adversarial labels. Fig. 3a shows when the dataset is lightly poisoned, the effect of NN is not obvious, and the total poison rate in the top 30% of the data is similar when using or not using NN. But, when the poison rate is high, NN contributes significantly to decreasing the cosine similarities of the poisoned pairs, marking a 7.3% drop in total poison rate in the top 30% data.

## 5.3 IMPORTANCE OF SAFECLIP MIXED TRAINING

SAFECLIP's mixed training epochs gradually incorporate increasing amounts of data as safe, allowing them to be trained with the CLIP loss, while keeping poisoned pairs away throughout the training process. This is essential for both the high performance of SAFECLIP, despite initially being trained with a small amount of data, and for achieving a low attack success rate. Specifically, two design choices contribute significantly to our mixed training: (1) unimodal CL training on risky data. (2) the small increment ratio $i$. Here, we illustrate the necessity of each of these design choices. We run our experiments on 100K data from CC3M. We randomly select 16 different images as our target images, and pair each with 30 adversarial captions. In total, 480 poisoned pairs are injected into the dataset.

**Unimodal CL on risky data** Similar to unimodal warmup epochs as discussed in Sec. 4.1.1, training with unimodal CL on risky data can improve SAFECLIP's defense effectiveness significantly. As shown in Fig. 3c, unimodal CL helps on *risky* data decrease the PSR by 12.5%.

**Small increment ratio** Larger increment ratio for updating *safe* data improves CLIP's performance, as it allows more data to be trained with CLIP loss earlier. However, this can put the model into risk if any poisoned pairs enter the safe data. As shown in Fig. 4, when we choose a larger $i = 5$, the PSR is increased by 37.5%.

## 6 CONCLUSION

We proposed SAFECLIP, an effective method for safely pre-train CLIP against targeted data poisoning and backdoor attacks. Using inmodal CL warmup and slow-paced CLIP warmup, SAFECLIP filters out majority of the poisons before pre-training and continue to exclude them during pre-training with mixed training strategy. Through extensive experiments, we demonstrated that SAFECLIP drops the attack success rate down to 0% for backdoor attacks and 6.25 % for targeted data poisoning attack with poisoning ratio of 0.05%, while preserving the CLIP performances on various downstream datasets.

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

# 7 APPENDIX

## 7.1 BENCHMARK DATASETS

To evaluate the downstream performance of our model, we conduct linear probe and zero-shot classifications, as introduced in Sec. 3.1, on 10 widely used datasets (Radford et al., 2021; Li et al., 2021; Yang & Mirzasoleiman, 2023) listed in Table 4.

Table 4: Details of downstream datasets.

| Dataset | Classes | Train Size | Test Size |
|---|---|---|---|
| CIFAR10 | 10 | 50,000 | 10,000 |
| CIFAR100 | 100 | 50,000 | 10,000 |
| Food-101 | 101 | 75,750 | 25,250 |
| DTD | 47 | 3,760 | 1,880 |
| FGVC Aircraft | 100 | 6,667 | 3,333 |
| Flowers-102 | 102 | 2,040 | 6,149 |
| Caltech-101 | 102 | 3,060 | 6,085 |
| OxfordIIITPet | 37 | 3,680 | 3,669 |
| Stanford Cars | 196 | 8,144 | 8,041 |
| ImageNet1K | 1000 | 50,000 | 50,000 |

## 7.2 AUGMENTATION POLICY

For data augmentation in SAFECLIP, we used the SimCLR image augmentation method including random image cropping, horizontal flipping, color jittering, grayscale conversion, and blurring (Chen et al., 2020). For text modality, we used the same Easy Data Augmentation proposed in (Wei & Zou, 2019), which applies simple text token transformation like synonym replacement and random delete.

## 7.3 ADDITIONAL BACKDOOR ATTACKS

In this section, we investigate if SAFECLIP can defend a variety of other backdoor attacks including those with non-patch-like triggers. We consider three different backdoor attacks: Blended, WaNet, and Label-consistent attack. (Chen et al., 2017; Turner et al., 2019; Nguyen & Tran, 2021) For all three attacks, we used a higher poison rate of 0.015%, train on CC1M for 32 epochs. The result is shown in Tab. 5. As shown, SAFECLIP can defend a variety of backdoor attacks effectively.

Table 5: Effectiveness of SAFECLIP in defending non-patch-like backdoor attacks and label-consistent attack

| Model | ‖ | # Blended | BSR | ‖ | # WaNet | BSR | ‖ | # Label-Consis | BSR |
|---|---|---|---|---|---|---|---|---|---|
| CLIP | ‖ | 1500 | 99.3% | ‖ | 1500 | 96.3% | ‖ | 1500 | 71.0% |
| SAFECLIP | ‖ | 1500 | 0% | ‖ | 1500 | 0% | ‖ | 1500 | 0.03% |

## 7.4 TUNING SAFECLIP FOR DIFFERENT DATASET

In this section, we discuss how to apply SAFECLIP to datasets of varying sizes and distributions. Empirically, we found that we found that the learning rate of the CLIP epoch at the end of the warmup should be set so that the average cosine similarity between all image-caption pairs after the warmup is close to but does not exceed half of the average cosine similarity between all image-caption pairs after a regular CLIP epoch training with a standard learning rate. To confirm this, we conducted experiments on CC3M, CC1M, and MSCOCO datasets (Lin et al., 2014) comprising 80K image-caption pairs. CC and MSCOCO have very different data distributions. We included this dataset to demonstrate the effectiveness of our method across datasets with varying distributions and sizes. For all datasets, we train for 48 epochs. We maintained a consistent poison rate of 0.05% for all datasets and confined our training to the unimodal warmup and CLIP warmup phases. The results of these experiments are detailed in Table 6. As demonstrated, we lowered the learning rate for each dataset

to ensure that the average cosine similarity of all image-caption pairs following one epoch of CLIP warmup does not exceed half of that achieved after a standard CLIP epoch training with a regular learning rate. Our method is a simple and effective method for hyperparameter tuning for SafeCLIP, since none of the models were poisoned by the end of the training.

Table 6: Amount of remained poisons in the top 15% and 30% of the training dataset. By adjusting the learning rates such that the average cosine similarity of all the image-caption pairs after the warmup is at most half of the average cosine similarity after one CLIP training, we can ensure that our model will not be poisoned.

| Dataset | Adj.lr | Std.lr | Warmup avg sim | CLIP avg sim | top 15% PSR | top 30% PSR | Final PSR |
|---------|--------|--------|----------------|--------------|-------------|-------------|-----------|
| CC3M | 5e-6 | 5e-4 | 0.17 | 0.37 | 0.1% | 1.5% | 6.25% |
| CC1M | 5e-6 | 5e-4 | 0.13 | 0.37 | 0.8% | 4.0% | 6.25% |
| COCO | 5e-5 | 5e-4 | 0.21 | 0.43 | 0% | 2.5% | 0% |

## 7.5 SAFECLIP ON DIFFICULT EXAMPLES

In this section, we discussed the potential influence of SAFECLIP on training on complex data. As SAFECLIP divides the dataset into risky set and safe set based on cosine similarities, it is natural to raise concern about if hard clean image-caption pairs will be classified into the risky set and trained with only unimodal loss as well. However, this is not the case in general: If we extend the training period to e.g. 64 epochs, close to 80% of the data will end up in the safe set and trained with CLIP loss, and many of the complex examples will be included in the safe set as well. Note that, since we gradually increase the size of the safe set, SAFECLIP can successfully prevent including poisoned pairs into the safe set, and does not train on them with the CLIP loss. Hence, it can successfully defend the model. The following table shows the result of training SAFECLIP on CC3M for 64 epochs instead of 32 epochs. For both of the epoch numbers, the models are barely poisoned, but the performance is much higher. This can empirically show that more data, including those that are more difficult, are included in the CLIP training dataset, leading to the performance gain.

Table 7: By training for more epochs, SAFECLIP can incorporate more complex image-caption pairs into the safe set, thus increasing the performance of the model, without harming the defense.

| Dataset | Zero-shot | | | Linear-probe | | | PSR | BSR |
|---------|-----------|------|------|--------------|-------|-------|-----|-----|
| | C10 | C100 | I1K | C10 | C100 | I1K | | |
| SafeCLIP-32 | 39.7 | 10.41 | 9.87 | 71.9 | 47.32 | 24.53 | 6.25% | 0% |
| SafeCLIP-64 | 43.05 | 14.39 | 12.57 | 75.02 | 50.62 | 28.65 | 6.25% | 0% |

**Limitation.** SafeCLIP warms up the model with in-modality CL followed by 1 CLIP epoch with small learning rate to distinguish the clean and poisoned pairs. However, if the number of injected poisons are too high, SafeCLIP may not be able to distinguish the poisoned pairs from the clean pairs. From our experiments, we were not able to effectively distinguish the majority of poisoned pairs after warmup, when poison rate is as high as 0.5%. If a small clean dataset of image-caption pairs is available, SafeCLIP can leverage that to defend a much higher poison rate.

