# OpenReview forum: "Better Safe than Sorry: Pre-training CLIP against Targeted Data Poisoning and Backdoor Attacks"
_ICLR.cc/2024/Conference — Submitted to ICLR 2024_

### Official Review · Reviewer_2rqV · 2023-10-14

**Soundness:** 3 good
**Presentation:** 2 fair
**Contribution:** 2 fair
**Rating:** 3
**Confidence:** 4

**Summary:**

This paper studies the defence against data poisoning and backdoor attacks on Contrastive Language-Image Pre-training (CLIP). It proposed a method called SafeCLIP based on the observation of the following: 1) the backdoored image-text pairs are learned first. 2) unimodal learning can break the backdoor connection between image and text pairs. The method first performs unimodal learning using the NNCLR method, then splits the data into safe and risky subsets. The CLIP is performed on the safe set, while the unimodal learning is performed on the risky set. The results show that SafeCLIP is effective against existing attacks.

**Strengths:**

- The observation of using a small learning rate with CLIP for an epoch can show a distinctive difference between poison and clean pairs, which is new and interesting.
- Using NNCLR to cluster backdoor images is interesting and technically sound for "warming up" the model against a visible patch as the trigger.

**Weaknesses:**

- The NNCLR as unimodal training might not effectively cluster other (non-patch-like) backdoor triggers, and it will be more convincing if the proposed method could demonstrate effectiveness on these triggers. Such as DFST [1], smoothed frequency domain trigger [2], or invisible trigger [3].
- The proposed method has a few components, whereas, in practice, it might not be that easy to get all hyperparameters in the suitable range. For example, in the experiments, the proposed method sample 1M out of CC3M and use 1 epoch with a learning rate for warmup. What if using all 3M? or on the CC12M dataset? Or even a large dataset? Is 1 epoch still apporiate? One might use the number of optimization steps as the threshold. However, it is not guaranteed it will encounter sufficient backdoor pairs within the given threshold. I don't think this metric is a robust and general way to separate the safe set and risky set.
- The "safe" incremental $i$% might be different as well for larger datasets.
- Section 4.1.1, "... to poison an image of a cat with a plane caption, the image ...", seems to be true for the dirty-label attack. What happens if it is a label-consistent attack?


[1] Cheng, Siyuan, et al. "Deep feature space trojan attack of neural networks by controlled detoxification." AAAI 2021.\
[2] Zeng, Yi, et al" "Rethinking the backdoor attacks' triggers: A frequency perspective." ICCV. 2021.\
[3] Li, Yuezun, et al. "Invisibl" backdoor attack with saattacks'cific triggers." ICCV 2021.

**Questions:**

- There seems to be a gap between the reported zero-shot accuracy on clean data and the reported by CleanCLIP for ImageNet. The results show 9.87% where, as in CleanCLIP, it is near 20\%. Is there a reason for this difference?
- It could be better to report the area under the ROC curve and precision-recall curve to distinguish the poison and clean pairs, e.g. the scenario in Fig 2. (b).

---

> ### Author Response · Authors · 2023-11-23
>
> We thank the reviewer for acknowledging the innovative aspect and soundness of our method.
>  - 1&4. Regarding SafeCLIP’s defense against non-patch-like backdoored triggers, We presented our method’s performance in defending different backdoor attacks: We implemented two additional non-patch-like backdoor attacks [1], [2], in addition to label-consistent attacks [3]. We trained our model with CC1M for 32 epochs. For all three attacks, we used a higher poison rate (0.15% as opposed to 0.05% in our paper) to attack the CLIP models.
>    - As shown, SafeCLIP is still effective against non-patch-like triggers, even with higher poison rate.
>
> | Attacks | Blended | WaNet | Label-Consis |
> |-----------|---------|-------|--------------|
> | CLIP      | 99.3%   | 96.3% | 71%          |
> | SafeCLIP  | 0%      | 0%    | 0.03%        |
>
> 2. If one epoch is an appropriate metric to separate the safe set and risky set.
>  - First, we argue that one epoch of CLIP with a low learning rate is generally required for different data distribution and dataset size. The primary objective of the CLIP warmup is to minimally train the model, to minimally associate **all** image-caption pairs. Training on only a subset of the dataset results in low cosine similarities for the remaining pairs. This makes the clean pairs that are not trained on with CLIP loss indistinguishable from the poisoned pairs (which also exhibit low cosine similarity).
>  - Indeed, to apply SafeCLIP on datasets of varying sizes and distributions, adjusting the (small) learning rate when applying one epoch of CLIP loss is sufficient: empirically, we found that if the learning rate of the CLIP epoch at the end of the warmup is such that the average cosine similarity between all image-caption pairs after the warmup is close to but does not exceed half of the average cosine similarity between all image-caption pairs after one CLIP epoch with standard learning rate, majority of the poisons end up in the risky set and SafeCLIP will be successful.
>  - To confirm this, we conducted new experiments on CC3M, CC1M, and MSCOCO datasets [6] comprising 80K image-caption pairs. CC and MSCOCO have very different data distributions. We included this dataset to demonstrate the effectiveness of our method across datasets with varying distributions and sizes. For all datasets, we train for 48 epochs (we will discuss this choice in the next part of our answer). We maintained a consistent poison rate of 0.05% for all datasets and applied SafeCLIP warmup to the poisoned datasets. The results of these experiments are detailed below.
> | dataset | Adj.lr | Std.lr | Warmup avg sim | CLIP avg sim | top 15% PR | top 30% PR | Final PSR |
> |---------|--------|--------|----------------|--------------|--------------|-------------|-----------|
> | CC3M    | 5e-6   | 5e-4   | 0.17           | 0.37         | 0.1%         | 1.5%        | 6.25%     |
> | CC1M    | 5e-6   | 5e-4   | 0.13           | 0.37         | 0.8%         | 4.0%        | 6.25%     |
> | COCO    | 5e-5   | 5e-4   | 0.21           | 0.43         | 0%           | 2.5%        | 0%        |
>   - As demonstrated, we lowered the learning rate for each dataset to ensure that the average cosine similarity of all image-caption pairs following one epoch of CLIP training does not exceed half of that achieved after one CLIP training epoch with standard learning rate. We see that this method is a simple and effective method for hyperparameter tuning for SafeCLIP, since none of the models were poisoned by the end of the training. We have added this discussion to Appendix 7.4.
>
> 3. If 1% incremental ratio is generalizable to larger datasets.
>  - We note that 1% incremental ratio is a very conservative increment ratio and is chosen to ensure that SafeCLIP can defend stronger attacks. In most situations, SafeCLIP can increase the number of examples in the safe set much faster (i.e. by larger increments) without poisoning the model. However, to be on the safe side, we used 1% without tuning this parameter, and we expect this to work across datasets and poisoning attacks. For example in the setting of the above experiment, after the warmup, more than 94% of the poisoned pairs are in the bottom 50% of the training data, so a higher incremental ratio can be utilized if needed.
>
> Q1. The reason for the performance gap is because we pretrained our model with 1M data with a batch size of 512. The reason for this choice was to be consistent with the setting considered in our only existing baseline, namely RoCLIP, which conducted experiments on CC 1M with 32 epochs of training. On the other hand, CleanCLIP used a pretrained CLIP from [5] on 3M dataset.

---

> ### Author Response · Authors · 2023-11-23
>
> - To further confirm that our results hold in more general settings, we re-run our main experiment on CC 3M with the batch size of 512, and present the result below. We also include the CLIP performance from [4] as a reference. **Note that, [4] train for 64 epochs with a batch size of 128. To match their performance, we train our model for 48 epochs.**
> - As shown, in this setting, our baseline CLIP matches the standard CLIP performance on 3M data. With 32 epochs, SafeCLIP has some performance deficits compared to CLIP. However, note that SafeCLIP also trained with a smaller amount of data with the small incremental ratio. We also presented CLIP and SafeCLIP trained with 48 epochs. Note that, there is not much performance gain for CLIP, but SafeCLIP’s performance increases significantly. Note that, continuing to train for more epochs does not compromise the model’s defense, as most of the poisons have very low cosine similarity and will not be selected to the safety set. Indeed, the poison rate at epoch 48 is still 6.25%.
> |  Dataset   | C10 ZS | C100 ZS | I1K  ZS | C10 LP | C100 LP | I1K LP |
> |----------|-----------|------|------|--------------|------|------|
> | CLIP-aug | 29.73     | 9.02 | 15.26| 80.00        | 58.25| 35.34|
> | SafeCLIP | 43.05     | 14.39| 12.57| 75.02        | 50.62| 28.65|
>
> Q2. Thanks for the suggestion. We will include both these figures in our revision.
>
> [1] Xinyun Chen, Chang Liu, Bo Li, Kimberly Lu, and Dawn Song. Targeted backdoor attacks on deep learning systems using data poisoning. arXiv preprint arXiv:1712.05526, 2017.
>
> [2] Tuan Anh Nguyen and Anh Tuan Tran. Wanet - imperceptible warping-based backdoor attack. In International Conference on Learning Representations, 2021.
>
> [3] Alexander Turner, Dimitris Tsipras, and Aleksander Madry. Label-consistent backdoor attacks. arXiv preprint arXiv:1912.02771, 2019.
>
> [4] Goel, Shashank, et al. "Cyclip: Cyclic contrastive language-image pretraining." Advances in Neural Information Processing Systems 35 (2022): 6704-6719.
>
> [5] Cherti, Mehdi, et al. "Reproducible scaling laws for contrastive language-image learning." Proceedings of the IEEE/CVF Conference on Computer Vision and Pattern Recognition. 2023.

---

### Official Review · Reviewer_MgX1 · 2023-10-17

**Soundness:** 4 excellent
**Presentation:** 3 good
**Contribution:** 3 good
**Rating:** 8
**Confidence:** 4

**Summary:**

The paper introduces a novel defense against data poisoning and backdoor attacks in multi-modal contrastive learning settings like CLIP. The approach consists of two stages, an unimodal self-supervised warm-up stage and a training stage. In the first stage, both encoders (image + text) are trained separately in a self-supervised manner to learn meaningful representations of inputs without the components being affected by the poisoned sample pairs. Instead of solely applying standard contrastive learning methods, the sample matching in the embedding space is done with nearest neighbors from a sample pool to avoid cluster building of poisoned samples. After both components are trained, the whole system is trained with the standard CLIP objective for a single epoch and a low learning rate to connect similar image and text embeddings. In the second stage, the training data is divided into safe and risky data based on the assigned cosine similarity by the system itself. Both sets are updated after each training epoch with the tendency to increase the number of safe samples. Whereas training with safe samples also uses the standard CLIP objective, the risky samples are again used with unimodal contrastive learning. The evaluation compares the proposed defense method with two existing approaches and states significant improvements in terms of utility and robustness.

**Strengths:**

- The paper addresses an important topic of model security and proposes an interesting and novel defense mechanism. Since the approach does not require a clean model or a clean subset of samples, it offers a more practical defense strategy compared to existing work.
- While the evaluation is only done on one dataset (CC3M) and one CLIP architecture (ResNet+Transformer), the results are convincing and demonstrate the high effectiveness of the defense mechanism. I particularly like the comprehensive ablation study to illustrate the impact and requirement of each component of the defense strategy.
- The paper is well-written and easy to understand. All parts of the approach are clearly motivated and described.

**Weaknesses:**

- Dividing samples into clean and risky sets by the assigned cosine similarity seems reasonable. However, I wonder if this also affects "hard", i.e., complex samples, and tends to put them into the risky set even if those are not poisoned in any way. I think this might harm the model's performance on more complex tasks. Since the evaluation is only done on large common benchmarks, they do not account for the model's predictive abilities on unseen hard samples.
- Also, only measuring the attack success rate might not be enough to evaluate the backdoor effectiveness. If the samples containing triggers are not predicted as the attack's target class but also not as their ground-true class, the attack is still effective in the sense of an evasion attack. I think measuring if the samples containing triggers are also predicted as the ground-true class would be an important metric to demonstrate that the approach indeed overcomes the backdoor attacks.

Small remarks:
- I think section 5.2 is not only an "ablation study" but also includes a sensitivity analysis. Maybe this should be highlighted in the section title / section itself.
- I think citing Carlini and Terzis (Poisoning and backdooring contrastive learning, 2022, ICLR(!)) and Carlini et al. (Poisoning Web-Scale Training Datasets is Practical, 2023) in the introduction would further support the statements made by the paper.
- There are a few typos in the paper, e.g., "exaples" in 1 and a missing space in 5.2.2 "(2)the"

**Questions:**

- Would you expect the approach to also work for higher poisoning rates or different trigger patterns? Or is there to be an expected limit on the poisoning rate that the method can handle? To clarify, I would expect that the distinction between clean and risky samples might not work well if the number of poisoned samples is rather large.

---

> ### Comment · Reviewer_MgX1 · 2023-11-23
>
> Dear all,
>
> I would have loved to read and acknowledge a rebuttal statement by the authors. However, at least for me, there is no answer visible. I am not sure if the authors decided not to submit a statement or if there is any problem with the visibility.
>
> Best,
> Reviewer MgX1

---

> > ### Comment · Reviewer_2rqV · 2023-11-23
> >
> > Dear all,
> >
> > There is no answer visible from my end as well.
> >
> > Best,\
> > Reviewer 2rqV

---

> ### Author Response · Authors · 2023-11-23
>
> We thank the reviewer for acknowledging the significance of our topic, the practicality of our method, the persuasiveness of our argument, and the well-written nature of our paper.
>
> We apologize for not posting our responses earlier, as we were waiting for our experiments on CC 3M to finish.
>
> 1. We agree with the reviewer that dividing samples into clean and risky sets and applying CLIP loss only to safe set could potentially harm the complex data. However, note that  during the training the safe set is gradually getting larger while the risky set is getting smaller. In our original experiment, where we trained for 32 epochs (to be consistent with our RoCLIP baseline), in total 47% of examples ended up in the safe set, and 53% of examples ended up in the risky set. However, if we extend the training period to e.g. 64 epochs, close to 80% of the data ends up in the safe set and trained with CLIP loss, and many of the complex examples will be included in the safe set as well. Note that, since we gradually increase the size of the safe set, SafeCLIP can successfully prevent including poisoned pairs into the safe set, and does not train on them with the CLIP loss. Hence, it can successfully defend the model. The following table shows the result of training SafeCLIP on CC3M for 64 epochs instead of 32 epochs. For both of the epoch numbers, the models are barely poisoned, but the performance is much higher. This can empirically show that more data, including those that are more difficult, are included in the CLIP training dataset, leading to the performance gain. We have added this discussion in Appendix 7.5.
>
> 2. We thank the reviewer for the suggestions. To show that SafeCLIP can correctly predict the backdoored patched images, we compared the zero-shot performance of backdoored SafeCLIP trained on CC 3M on regular ImageNet1K test set versus its performance on patched ImageNet1K test set. In accordance with our backdoor attacks, we patched all the test set images on the left side corners. For regular ImageNet1K, SafeCLIP achieves a zero-shot performance of 18.16%, while on patched ImageNet1K test set, SafeCLIP achieves a zero-shot performance of 14.12%. While the performance is slightly lower, SafeCLIP can still produce high-quality mode and representations for the data. We will add the evaluation with this metric for our method and the RoCLIP baseline to our revised version.
>
> |  Dataset | C10 ZS | C100  ZS| I1K ZS| C10 LP | C100 LP | I1K LP | PSR|  BSR |
> |------------|-----------|------|------|--------------|------|------|-------|-----|
> | SafeCLIP: 32 epochs| 39.7      | 10.41| 9.87 | 71.9         | 47.32| 24.53| 6.25% | 0%  |
> | SafeCLIP: 64 epochs| 43.05     | 14.39| 12.57| 75.02        | 50.62| 28.65| 6.25% | 0%  |
>
> 3. We thank the reviewer for the remarks on our ablation study, our citation, as well as our typos. We have incorporated these modifications into our manuscript.
>
> 4. We presented our method’s performance in defending different backdoor attacks: We implemented two additional non-patch-like backdoor attacks [1], [2], in addition to label-consistent attacks [3]. We trained our model with CC1M for 32 epochs. For all three attacks, we used a higher poison rate (0.15% as opposed to 0.05% in our paper) to attack the CLIP models.
>
> | Attacks | Blended | WaNet | Label-Consis |
> |-----------|---------|-------|--------------|
> | CLIP      | 99.3%   | 96.3% | 71%          |
> | SafeCLIP  | 0%      | 0%    | 0.03%        |
>
> - As shown, SafeCLIP is still effective against non-patch-like triggers, even with higher poison rate.
> - As for the general limit of our model: SafeCLIP warms up the model with in-modality CL followed by 1 CLIP epoch with small learning rate to distinguish the clean and poisoned pairs. However, if the number of poisons are too high, SafeCLIP may not be able to distinguish the poisoned pairs from the clean pairs. From our experiments, we were not able to effectively distinguish the majority of poisoned pairs after warmup, when poison rate is as high as 0.5%. If a small clean dataset of image-caption pairs is available, SafeCLIP can leverage that to defend a much higher poison rate. We have included this discussion in our revised Appendix. Thanks for your suggestion!
>
> [1] Xinyun Chen, Chang Liu, Bo Li, Kimberly Lu, and Dawn Song. Targeted backdoor attacks on deep learning systems using data poisoning. arXiv preprint arXiv:1712.05526, 2017.
>
> [2] Tuan Anh Nguyen and Anh Tuan Tran. Wanet - imperceptible warping-based backdoor attack. In International Conference on Learning Representations, 2021.
>
> [3] Alexander Turner, Dimitris Tsipras, and Aleksander Madry. Label-consistent backdoor attacks. arXiv preprint arXiv:1912.02771, 2019.

---

### Official Review · Reviewer_TTCu · 2023-11-03

**Soundness:** 1 poor
**Presentation:** 2 fair
**Contribution:** 2 fair
**Rating:** 3
**Confidence:** 5

**Summary:**

Contrastive Language-Image Pre-training (CLIP) is an approach to learn a joint embedding space for images and languages with image-caption data. This paper proposes a new CLIP-based pre-training strategy called SafeCLIP to make it more robust against data poisoning attacks and backdoor attacks.


They first apply unimodal contrastive learning to pre-train the image encoder and text encoder separately.
Then they apply the original CLIP training using all data for 1 epoch with a smaller learning rate.
Assuming the models are not yet effectively poisoned/backdoored, they then use the similarity between the current image embeddings and caption embeddings as indicators of whether the training data are poisoned/backdoored for later training, where they apply regular CLIP loss on samples with high similarity between the image embeddings and text embeddings and apply unimodal contrastive learning to the others.

**Strengths:**

1. The subject is important. Given the impact of CLIP, how to make it robust against data poisoning/backdoor attacks can be interesting to many from both research and application communities.

**Weaknesses:**

1. Key assumptions for the proposed defense are flawed and unverified.

There are two major (necessary but not sufficient) assumptions for the proposed method to be effective: 1) Contrastive learning on individual modalities (image/text) is immune to poisoning/backdoor attacks. 2) One epoch of CLIP training with reduced learning rate over potentially poisoned/backdoored data is safe.

Regarding assumption 1), it may be true for some attacks designed specifically for CLIP (i.e. they inject malicious patterns through the image-caption pairs). However, given that we have not only backdoor attacks on (unimodal) self-supervised learning [1] and also poisoning attacks utilizing the unlabeled part of (unimodal) semi-supervised learning [2], the assumption does not seem to be true in general, which is a critical issue.

Assumption 2) is conceptually fine but the execution is not satisfying. Specifically, from Table 3 we see that the effectiveness of filtering is sensitive to many design choices (e.g. how many unimodal contrastive learning & how many CLIP training with lowered learning rate). Note that '1 epoch' is likely not a good, universal measurement of how many slow-paced CLIP training should be conducted: If one has a new set of data, with difference distributions and different number of samples as used in yours, how can one decide these parameters when they do not know the poisoned/backdoor samples?


2. The reported performance numbers (for both baseline and the proposed method) seem a bit low. I understand the numbers should be much lower than the actual CLIP since the authors have to use (a part of) a smaller dataset. However, '32 epochs' for 1 million samples seems insufficient (which is insufficient to train supervised models on ImageNet, unless with carefully designed training schemes, or for most if not all unimodal contrastive learning methods). The original CLIP paper indeed trains only 32 epochs but they are using much more data than yours. These raise concerns regarding the usefulness of these results.

3. In addition, some of the performance boosts are attributed to the 'data augmentation' (i.e. line 10 of Algorithm 1). I am not able to find the details of these augmentations and if I understand correctly, these augmentations are not used in your implementation of the baseline CLIP. This does not seem to be a fair comparison unless good reasons are provided to include these augmentations as part of the contributions.


**references:**

[1] Saha, A., Tejankar, A., Koohpayegani, S.A. and Pirsiavash, H., 2022. Backdoor attacks on self-supervised learning. In Proceedings of the IEEE/CVF Conference on Computer Vision and Pattern Recognition (pp. 13337-13346).

[2] Carlini, N., 2021. Poisoning the unlabeled dataset of {Semi-Supervised} learning. In 30th USENIX Security Symposium (USENIX Security 21) (pp. 1577-1592).

**Questions:**

Please see the Weakness part above for detailed issues.

For 1: Regarding assumption 1), I think it is a fundamental issue and I would like to hear your response. For assumption 2), I want to hear if authors have proposals about how these design choices (e.g. the number of unimodal pretraining epochs, the number of slow-paced CLIP epochs) can be determined without knowing the poisoned/backdoor samples.

For 2: Empirical results justifying (i.e. showing that these hyper-params lead to a fairly good results given the dataset) the hyper-parameters for the **baseline implementation of CLIP** on your training set is necessary unless other evidence can be provided for justification.

For 3: Please include these augmentations in the baseline as well unless there is a good reason to include these augmentations as part of the contributions.

---

> ### Author Response · Authors · 2023-11-23
>
> We thank the reviewer for acknowledging the importance of our work in making CLIP more robust to data poisoning and backdoor attacks. Below we discuss each comment in detail.
> - First we discuss the two assumptions mentioned by the reviewer:
>   - *SafeCLIP requires contrastive learning on individual modalities to be immune against poisoning/backdoor attacks*.
>     - Indeed, SafeCLIP does not require this assumption to hold. Below we discuss attacks on contrastive learning and semi-supervised learning, respectively:
>       - CL Attacks: Attacks on Contrastive Learning (CL), such as [1], select a target category and patch some of its training images with a backdoor trigger. In doing so, CL clusters all images with the backdoor trigger together. When a linear classifier is trained on the representations, any image with the backdoor trigger is classified as the target category.
>       - Such attacks do not affect SafeCLIP:
>         - Assume we pair such images with captions related to their correct category. Otherwise, the attack becomes a standard backdoor attack on CLIP which we have already considered in our paper. First, note that if the backdoored image-caption pairs are not trained on with the CLIP loss, the model does not associate the backdoored images to the corresponding text category. During training with SafeCLIP, images containing the backdoor patch end up in the risky set and are not trained on with the CLIP loss. This is because the backdoor patch makes such images dissimilar to the rest of the images in their category. Hence, as long as the backdoor rate is not too high (standard backdoor rate when training CLIP is very small~0.05%), the backdoored images do not get close enough to the adversarial text category during warmup and later during the training. Hence, SafeCLIP will not be affected.
>         - Moreover, note that using the Nearest Neighbor (NN) pool and data augmentation during SafeCLIP unimodal CL are effective strategies against backdoor attacks on unimodal CL. In particular, when the fraction of backdoored images is not too large (standard backdoor rate when training CLIP is very small~0.05%), the NN pool and data augmentation effectively prevent the backdoored images from clustering in the representation space. Thus, the backdoored images cannot effectively cluster in the image representation space.
>         - As the backdoored images do not end up in the safe set and do not cluster tightly in the image representation space, they cannot poison the model (zero-shot or linear-probe evaluation) when SafeCLIP is employed.
>         - To confirm the above argument, we conducted an experiment on the 1M CC dataset, with the batch size of 512 and 32 epochs of training (we will discuss this choice of epochs in the next part of our rebuttal). To apply [1] to CLIP, we patched the triggers on random images from a randomly chosen target category, and paired such images with captions related to the target category (note that pairing images with captions related to another category falls under standard backdoor attacks on CLIP, which we have already studied in our paper). In addition, even though prior work considered a maximum of 0.05% backdoor rate, we increased the poison rate to 0.15% to strengthen the backdoor attack. After the warmup, 94.5% of the backdoored images end up in the “risky” set and trained on with unimodal CL, so we consider this a successful and effective attack attempt against SafeCLIP’s unimodal structure.
>         - We evaluated the model’s defense with both zero-shot and linear-probe classification. Notably, at the end of training, CLIP was poisoned with a 71% backdoor success rate (BSR), while SafeCLIP was barely poisoned with a 0.33% BSR. When evaluating the SafeCLIP model with linear prob, BSR is also 0%.
>    - Semi-SL attacks: attacks on semi-supervised learning, such as [2], select a target category, a labeled image ‘a’ in the target category, and a target unlabaled image ‘b’ from another category. Then, the adversary injects unlabeled images interpolating between image ‘a’ and image ‘b’, such that unlabeled image ‘a’ is classified as the target category.
>      - To apply this attack to CLIP, we need to first assign captions to the generated poisoned images that interpolate between a and b. In particular, such captions should be related to the target category for the attack to have the desired effect.
>      - Note that around half of the interpolated images are more similar to ‘b’ which is from a category other than the target, but are paired with captions related to the target category. Such pairs act as (weaker) targeted data poisoning attacks on CLIP, which we already considered in our paper. Such pairs will have a low cosine similarity between their image-caption pairs, and end up in the risky set. Hence, SafeClip can effectively defend the model against such attacks. The remaining poisoned images do not poison the model, as they have correct matching image-caption pairs.

---

> ### Author Response · Authors · 2023-11-23
>
> - Similar to the above scenarios, other attacks on CL or SSL will not be effective when training with CLIP or SafeCLIP. We also note that:
>   - Attacks on CLIP require to disrupt the association between image-caption pairs to affect the zero-shot performance of the model. If such attacks pair the poisons with wrong captions, SafeCLIP effectively breaks the attack. If they pair the poisons with correct captions, as long as PSR is not too large, they won't affect SafeCLIP. Using the nearest neighbor pool and data augmentation further improve the robustness of CL against attacks on CL and semi-SL.
>   - Unimodal SSL or CL attacks require 1% poison rate for the attack to succeed. Note that CLIP data is often much larger, and 1% of the dataset translates to a very large amount of poisons, which is often not feasible to generate and inject into the dataset.
>   - Finally, we note that any defense mechanism designed for CL can be applied together with SafeCLIP.
>
> - Next, we discuss the second assumption mentioned by the reviewer:
>   - *one epoch of CLIP training with reduced learning rate over potentially poisoned/backdoored data is safe*.
> - First,  we argue that one epoch of CLIP with a low learning rate is generally required for different data distribution and dataset size. The primary objective of the CLIP warmup is to minimally train the model, to minimally associate **all** image-caption pairs. Training on only a subset of the dataset results in low cosine similarities for the remaining pairs. This makes the clean pairs that are not trained on with CLIP loss indistinguishable from the poisoned pairs (which also exhibit low cosine similarity).
> - Indeed, to apply SafeCLIP on datasets of varying sizes and distributions, adjusting the (small) learning rate when applying one epoch of CLIP loss is sufficient: empirically, we found that if the learning rate of the CLIP epoch at the end of the warmup is such that the average cosine similarity between all image-caption pairs after the warmup is close to but does not exceed half of the average cosine similarity between all image-caption pairs after one CLIP epoch with standard learning rate, majority of the poisons end up in the risky set and SafeCLIP will be successful.
> - To confirm this, we conducted new experiments on CC3M, CC1M, and MSCOCO datasets [6] comprising 80K image-caption pairs. CC and MSCOCO have very different data distributions. We included this dataset to demonstrate the effectiveness of our method across datasets with varying distributions and sizes. For all datasets, we train for 48 epochs (we will discuss this choice in the next part of our answer). We maintained a consistent poison rate of 0.05% for all datasets and applied SafeCLIP warmup to the poisoned datasets. The results of these experiments are detailed below.
> | dataset | Adj.lr | Std.lr | Warmup avg sim | CLIP avg sim | top 15% PR | top 30% PR | Final PSR |
> |---------|--------|--------|----------------|--------------|--------------|-------------|-----------|
> | CC3M    | 5e-6   | 5e-4   | 0.17           | 0.37         | 0.1%         | 1.5%        | 6.25%     |
> | CC1M    | 5e-6   | 5e-4   | 0.13           | 0.37         | 0.8%         | 4.0%        | 6.25%     |
> | COCO    | 5e-5   | 5e-4   | 0.21           | 0.43         | 0%           | 2.5%        | 0%        |
> - As demonstrated, we lowered the learning rate for each dataset to ensure that the average cosine similarity of all image-caption pairs following one epoch of CLIP training does not exceed half of that achieved after one CLIP training epoch with standard learning rate. We see that this method is a simple and effective method for hyperparameter tuning for SafeCLIP, since none of the models were poisoned by the end of the training. We have added this discussion to Appendix 7.4.
>
> 2. Regarding our performance numbers on CC 1M and 32 epochs:
> The reason for this choice was to be consistent with the setting considered in our only existing baseline, namely RoCLIP, which conducted experiments on CC 1M with 32 epochs of training.
>  - The reason for this choice was to be consistent with the setting considered in our only existing baseline, namely RoCLIP, which conducted experiments on CC 1M with 32 epochs of training.
>  - To confirm that our results hold in more general settings, we re-run our main experiment on CC 3M with the batch size of 512, and present the result below. We also include the CLIP performance from [3] as a reference. **Note that, [3] train for 64 epochs with a batch size of 128. To match their performance, we train our model for 48 epochs.**
>
> | Dataset | C10  ZS| C100 ZS | I1K  ZS| C10 LP  | C100 LP | I1K  LP | PSR (0.05%) | BSR (0.05%)|
> |---------|-----|------|-----|-----|------|-----|-----|-----|
> | CLIP [3]   | 46.54 | 18.69 | 20.03 | 78.26 | 54.85 | 35.93 | - | - |
> | CLIP    | 48.19 | 18.58 | 19.40 | 77.81 | 54.72 | 33.17 | 100% | 30% |
> | SafeCLIP | 54.67 | 22.56 | 18.16 | 78.31 | 54.15 | 34.33 | 6.25% | 0% |

---

> ### Author Response · Authors · 2023-11-23
>
> - As shown, in this setting, our implementation of CLIP training matches the implementation of [3] on 3M data. In addition, we see that SafeCLIP’s defense is not influenced by the dataset size and yields the same 6.25% as reported in our paper. This confirms the strong defense capability of SafeCLIP while maintaining the CLIP performance. We have modified the performance number in the main paper accordingly.
>
> 3. Regarding the influence of data augmentation, we present new experimental results on CLIP with data augmentation and SafeCLIP. Due to time and computational limits, we trained both models on 1M data for 64 epochs. Compared to CLIP with data augmentation, SafeCLIP has a higher zero-shot performance, due to the unimodal CL training, while its linear-probe performance is lower. This is because we’re training on a much smaller number of (safe) image-caption pairs with CLIP loss, which is essential for breaking the attacks.
>
> | Dataset| C10 ZS | C100 ZS | I1K  ZS | C10  LP | C100 LP | I1K LP |
> |----------|-----------|------|------|--------------|------|------|
> | CLIP-aug | 29.73     | 9.02 | 15.26| 80.00        | 58.25| 35.34|
> | SafeCLIP | 43.05     | 14.39| 12.57| 75.02        | 50.62| 28.65|
> - For data augmentation, we used the SimCLR image augmentation method including random image cropping, horizontal flipping, color jittering, grayscale conversion, and blurring [4]. For text modality, we used the same Easy Data Augmentation proposed in [5], which applies simple text token transformation like synonym replacement and random delete. We added discussion of our augmentation policy in Appendix 7.2.
>
>
> [1] Kim, Minseon, Jihoon Tack, and Sung Ju Hwang. "Adversarial self-supervised contrastive learning." Advances in Neural Information Processing Systems 33 (2020): 2983-2994.
>
> [2] Jia, Jinyuan, Yupei Liu, and Neil Zhenqiang Gong. "Badencoder: Backdoor attacks to pre-trained encoders in self-supervised learning." 2022 IEEE Symposium on Security and Privacy (SP). IEEE, 2022.
>
> [3]Goel, Shashank, et al. "Cyclip: Cyclic contrastive language-image pretraining." Advances in Neural Information Processing Systems 35 (2022): 6704-6719.
>
> [4] Chen, T., Kornblith, S., Norouzi, M., and Hinton, G. A simple framework for contrastive learning of visual representations. In International conference on machine learning, pp. 1597–1607. PMLR, 2020.
>
> [5] Wei, J. and Zou, K. Eda: Easy data augmentation techniques for boosting performance on text classification tasks. arXiv preprint arXiv:1901.11196, 2019.
>
> [6] Lin, Tsung-Yi, et al. "Microsoft coco: Common objects in context." Computer Vision–ECCV 2014: 13th European Conference, Zurich, Switzerland, September 6-12, 2014, Proceedings, Part V 13. Springer International Publishing, 2014.

---

### Meta-Review · Area_Chair_dL33 · 2023-12-12

**Metareview:**

All reviewers discussed this paper after the rebuttal. None of the reviewers are positive about this paper. Reviewer MgX1 has a good rating of level 8, but in the discussion, Reviewer MgX1 agrees with the other reviewers and says "I am also fine to reject the paper". Hence, the paper will not be accepted. The main weaknesses are: lack of sufficient experiments, needs a lot of tuning in practice to get the correct hyperparameters, and the following assumptions; 1) Contrastive learning on individual modalities (image/text) is immune to poisoning/backdoor attacks. 2) One epoch of CLIP training with reduced learning rate over potentially poisoned/backdoored data is safe.

**Justification For Why Not Higher Score:**

The main weaknesses are: lack of sufficient experiments, needs a lot of tuning in practice to get the correct hyperparameters, and the following assumptions; 1) Contrastive learning on individual modalities (image/text) is immune to poisoning/backdoor attacks. 2) One epoch of CLIP training with reduced learning rate over potentially poisoned/backdoored data is safe.

**Justification For Why Not Lower Score:**

N/A

---

### Decision · Program_Chairs · 2024-01-16

Reject